# The importance of non-pharmaceutical interventions during the COVID-19 vaccine rollout

**Nicolò Gozzi**[1], **Paolo Bajardi**[2]\*, **Nicola Perra**[1]

**1** Networks and Urban Systems Centre, University of Greenwich, London, United Kingdom, **2** ISI Foundation, Turin, Italy

\* paolo.bajardi@isi.it

## Abstract

The promise of efficacious vaccines against SARS-CoV-2 is fulfilled and vaccination campaigns have started worldwide. However, the fight against the pandemic is far from over. Here, we propose an age-structured compartmental model to study the interplay of disease transmission, vaccines rollout, and behavioural dynamics. We investigate, via in-silico simulations, individual and societal behavioural changes, possibly induced by the start of the vaccination campaigns, and manifested as a relaxation in the adoption of non-pharmaceutical interventions. We explore different vaccination rollout speeds, prioritization strategies, vaccine efficacy, as well as multiple behavioural responses. We apply our model to six countries worldwide (Egypt, Peru, Serbia, Ukraine, Canada, and Italy), selected to sample diverse socio-demographic and socio-economic contexts. To isolate the effects of age-structures and contacts patterns from the particular pandemic history of each location, we first study the model considering the same hypothetical initial epidemic scenario in all countries. We then calibrate the model using real epidemiological and mobility data for the different countries. Our findings suggest that early relaxation of safe behaviours can jeopardize the benefits brought by the vaccine in the short term: a fast vaccine distribution and policies aimed at keeping high compliance of individual safe behaviours are key to mitigate disease resurgence.

**Data Availability Statement:** All data and code used to generate the results presented in the paper and in the supplementary information are available on Github (https://github.com/ngozzi/vaccine-

## Author summary

The start of vaccination campaigns is a decisive turning point in the global effort against COVID-19. Nonetheless, at least in the short and medium-term, vaccine availability and the logistical issues associated with an unprecedented mass vaccination suggest that non-pharmaceutical interventions will still play an important role in virus containment. Here, we propose an epidemic model to study the possible effects induced by a relaxation of COVID-safe behaviours in response to the vaccine rollout. Individuals may see this milestone as the end of the emergency and thus give up preventive measures potentially exposing themselves to higher infection risk. We explore the interplay between such behavioural changes and different population pyramids, contact patterns, epidemic conditions, vaccine allocation strategies, rollout speed, and vaccine efficacy. We show that early

behaviour) and Zenodo (https://doi.org/10.5281/zenodo.5266444).

**Funding:** NG acknowledges support from the DTA3/COFUND project funded by the European Union's Horizon 2020 research innovation programme under the Marie Skłodowska Curie grant agreement No 801604. PB acknowledges support from Intesa Sanpaolo Innovation Center. The funders had no role in study design, data collection and analysis, decision to publish, or preparation of the manuscript.

**Competing interests:** The authors have declared that no competing interests exist.

relaxation of COVID-safe behaviours can jeopardize and even nullify the benefit brought by the vaccine in the short and medium-term. Our results indicate that a high level of compliance to NPIs during vaccines rollout is crucial to avoid hindering the gigantic effort of the vaccination campaigns.

## Introduction

The COVID-19 pandemic has been largely fought with non-pharmaceutical interventions (NPIs). Bans of events and social gatherings, limitations in national and international travels, school closures, shifts towards remote working, curfews, closure of pubs and restaurants, cordon sanitaires, national and regional lockdowns are examples of governmental interventions implemented around the world to curb the spreading of SARS-CoV-2 [1–5]. While extremely effective, such top-down NPIs induce profound behavioural changes, bring many social activities to a halt, and thus have huge socio-economic costs. Hence, alongside these measures, governments nudged and/or mandated populations to adopt another set of NPIs. Social distancing, face masks, and increased hygiene are examples [6]. Although far from being cost-free, they are more feasible, sustainable, and allow for higher levels of socio-economic activity. As such, they have been the leitmotif of the post first COVID-19 wave in many countries. Unfortunately, awareness, adoption, and compliance with these NPIs have been spotty [6]. Furthermore, they have not been complemented with sufficiently aggressive test and trace programs. As result, many countries experienced marked disease resurgences after the summer 2020 and some had to resort to new lockdowns [7].

As we write, we have turned a crucial corner in hampering the resurgence, diffusion and severe outcomes of the disease. Several vaccines have shown great results of their phase 3 trials, and have been authorized for emergency use by several regulating agencies and tens of others are in the pipeline [8–10]. As result, we have witnessed the start of vaccination campaigns around the world. However, the logistical issues linked to the production, delivery, and administration of billions of doses on a global scale impose caution when evaluating the impact vaccines will have on the pandemic in the short and medium term. They will be a scarce resource and it will take time to vaccinate the fraction of the population necessary for herd immunity [11, 12]. Furthermore, vaccines are not perfect and there are many other unknowns [13]. The extent to which vaccines limit further transmission, how long the immunity lasts, and the levels of protection offered against new variants are key features currently under investigation [13]. So far, the results show high levels, around 95% [14, 15], of direct protection against the disease. Despite these first signs of the vaccine impact in the real world are extremely encouraging [16, 17], these figures might end up to be lower. Finally, vaccines' acceptance is a complex challenge. A recent survey among 13, 426 participants in 19 countries shows that, while 71.5% of the sample is very or somewhat likely to take the vaccine, there are large heterogeneities [18]. Acceptance rates vary from 90% in China to less than 55% in Russia. Furthermore, they are linked to socio-economic, socio-demographic features, and education attainment [18]. Arguably, vaccines alone will not be able to contain the spreading of the virus, at least in the short term [19, 20]. Social distancing, face masks, hygiene measures, and other NPIs will be still key during the delivery of such vaccination programs, especially considering the emergence of variants of concern as well as the challenges in vaccines procurement and administration in the developing world [21].

In this context, an important question emerges. What will happen to adoption and compliance to NPIs as vaccination campaigns progress? Their arrival and delivery might induce individual and collective behavioural changes. Some might see this milestone as the official end of

the emergency and as result relax their COVID-safe behaviours. Somehow paradoxically, vaccines might have, at least initially, a net negative effect. According to the health-belief model, one of the most commonly used psychological theories to characterize health-related behaviours, beliefs, perceptions, barriers to take action, and other modifying variables such as socio-demographic and socio-economic factors are key ingredients driving behavioural changes [22–24]. A recent study may offer an empirical evidence of this tendency [25]. By means of surveys delivered before the vaccination rollout in December 2020, authors found that vaccine information reduce adherence to social distancing, hygiene measures, and the willingness to stay at home. Several surveys conducted during the COVID-19 pandemic, well before any concrete hope for a vaccine, confirm this picture, provide hints of how the arrival of vaccines might corrode even more adoption, highlight how compliance is a complex multi-faced problem [5] and that risk-perception as well as NPIs adoption are indeed associated to several socio-economic determinants such as age, gender, wealth, urban-rural divide [26–41].

The literature aimed at estimating the epidemiological and societal impact of COVID-19 vaccines has been focused mainly on two very important points. The first line of research has been devoted to quantifying the effects of a vaccine on the evolution of the pandemic, considering different efficacy and coverage levels [42]. The second instead tackled the issue of vaccine allocation investigating strategies that target first different groups (i.e., age brackets, high-risk individuals) or particular occupations (i.e., doctors, nurses) [10, 43–45]. Very recently, the intuition that social distancing remains key during vaccination rollout stimulated few studies on the effects of a vaccine on the adoption of NPIs in specific settings [46–50]. An agent-based model applied to North Carolina showed that lifting NPIs during vaccines distribution would imply a substantial increase in infections and deaths [51]. Similarly, a data-driven model of SARS-CoV-2 transmission for China estimated that NPIs need to remain in place at least one year after the start of vaccination to avoid a generalized disease resurgence [52]. Despite these examples, the impact on real-world scenarios of the interplay of disease dynamics, behaviour change, and vaccines rollout is still largely unexplored. Furthermore, because of the novelty of the problem, also modeling frameworks able to characterize the possible behavioural dynamics linked to vaccinations are scarce.

To tackle such limitations, we introduce an age-structured compartmental epidemic model capturing the possible relaxation of NPIs adoption in response to the vaccine rollout. We model different compliance levels as distinct compartments and consider different behavioural dynamics driving the relaxation of NPIs. We test extensively the effects of behaviour change on disease spreading for different prioritization strategies, vaccine efficacy, and vaccination rollout speeds, using real demographic data and contacts matrices for six countries: Egypt, Peru, Serbia, Ukraine, Canada, and Italy. We choose these countries sampling levels of economic development. Indeed, in the World Economic Situation and Prospects 2020 issued by the United Nations, Egypt and Peru are classified as *Developing Economies* [53], Serbia and Ukraine as *Economies in Transition*, Canada and Italy as *Developed Economies*. Furthermore, considering dissimilar countries allows us to explore the interplay between vaccination and behaviours also as a function of population pyramids and intra/inter-generational mixing observed around the world. High income countries are typically characterized by higher average age, but lower inter-generational interactions respects to mid/low income countries [54]. These observations, together with the dependence of COVID-19 fatality rates on age, point to the possibility of non-trivial dependencies which we aim to explore here. As a way to realistically account for the different epidemic trajectories, we also explore the model after calibrating it on COVID-19 weekly deaths in the period 2020/09/01–2020/12/31 in these countries. To this end, we incorporate the timeline and effects of governmental restrictions on social contacts.

Our results provide quantitative insights on the interaction between sustained NPIs and an effective vaccination program. We show that an early relaxation of COVID-safe behaviours

may lower, and even nullify, the advantages brought by the vaccine in the short term. Overall, the picture that emerges from the analysis of the different countries is consistent: a high level of compliance towards NPIs such as mask-wearing, social distancing, and avoidance of large gatherings, is needed in order to avoid spoiling the great effort of the vaccination campaigns.

## Results

### Vaccine-behaviour model

We consider an age-structured epidemic model based on a *Susceptible-Latent-Infectious-Recovered* compartmentalization with the addition of pre-symptomatic and asymptomatic infection stages and deaths. On top of the disease dynamics, we model both the vaccination rollout and the behavioural responses linked to it. After the start of the vaccination rollout, on each day a fraction of the population receives a vaccine that decreases both the probability of infection (with efficacy $VE_S$) and of developing symptoms (with efficacy $VE_{Symp}$) and enters in the compartment $V$ [44]. As result, in our simulations, the overall efficacy of vaccine against severe outcomes such as death is $VE = 1 - (1 - VE_S)(1 - VE_{Symp})$. We introduce the rollout speed $r_V$ as the number of daily administered vaccine doses expressed as a percentage of the total population.

We study and compare three different strategies of vaccine prioritization. The first considers distributing vaccines in decreasing order of age. Previous research on COVID-19 vaccines allocation has shown that this strategy is the most effective in reducing the number of deaths and overall severity [10, 43, 44]. It is also the main strategy currently deployed in the different vaccination campaigns across the globe. In the second and third strategy vaccines are either distributed homogeneously or first to the individuals in age brackets 20–49 [55] and then to the rest of the population. The last strategy seeks to reduce symptomatic transmission targeting the most social active part of the population [10]. For simplicity, we will refer to the three rollout approaches as *vaccine strategy 1*, *vaccine strategy 2*, and *vaccine strategy 3*. In parallel to the vaccination, individuals (both susceptible $S$ and vaccinated $V$) may start giving up safe behaviours and expose themselves to higher infection risks. We introduce the compartments $S^{NC}$ and $V^{NC}$ for susceptible and vaccinated individuals respectively, to describe different behavioural classes (for convenience "NC" refers to "non-compliant" to COVID-safe behaviours). The increased infection risk for these individuals is captured by the parameter $r > 1$ (for example, $r = 1.3$ indicates an increased risk of 30%). Sensitivity analysis on parameter $r$ has been performed for values in the range of current estimates, as reported in Materials and methods section). We model the transition from $S$ and $V$ towards riskier behavioural classes as a function of the fraction of the vaccinated population and the parameter $\alpha$. In turn, we imagine that a worsening of the epidemiological conditions may push non-compliant individuals back to safer behaviours. Here, we consider the number of fatalities per 100, 000 individuals in the previous time step (i.e., day) and the parameter $\gamma$ to control the second behavioural transition. We refer the reader to the Materials and Methods section and the S1 Text for more details on the model. We refer to the the S1 Text, for results related to different behavioural mechanisms: a simpler one with constant rate, and a more complex one with different rates for vaccinated vs. unvaccinated individuals.

### Interplay between NPIs adoption and vaccination campaign

We use data for six different countries: Egypt, Peru, Serbia, Ukraine, Canada, and Italy. As mentioned above, these countries have been selected to sample a range of demographics and socio-economic contexts. In Fig 1A, we represent some key characteristics of the demographic and the mixing patterns between age groups of these countries. First, we consider the fraction of people aged over 65. This is a measure of the epidemic fragility of a country: indeed 65+ individuals are at particularly high risk of mortality from COVID-19 disease [56]. Italy is the

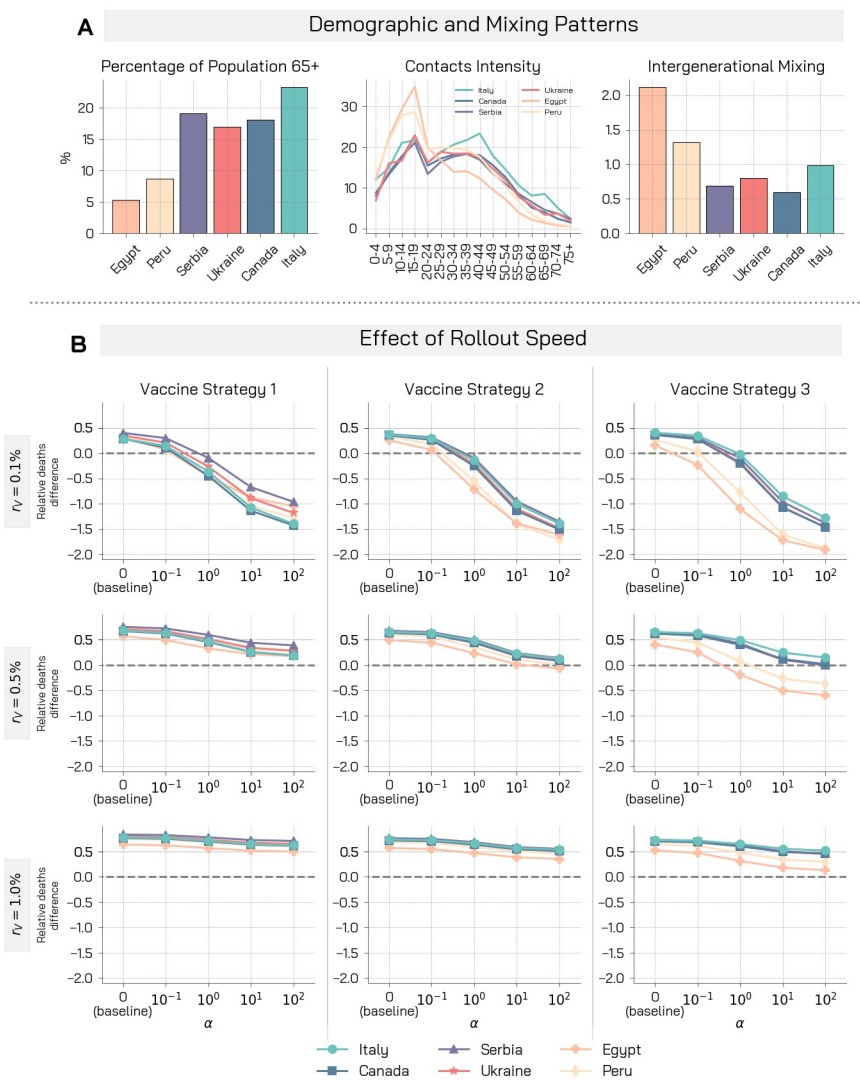

**Fig 1. Demographic, mixing patterns, and relative deaths difference for different rollout speeds and prioritization strategies.** A) Demographic characteristics and mixing patterns of the six countries considered are shown: percentage of the population aged over 65, contacts intensity for different age groups, and a measure of inter-generational mixing. B) Relative deaths difference is computed as the fraction of deaths that are avoided with a vaccine with respect to a baseline simulation without vaccine. We display results of the simulations for three vaccine rollout speed and prioritization strategies. Other parameters used are $\gamma = 0.5$, $R_0 = 1.15$, $r = 1.3$, $VE_S = 70\%$ and $VE_{Symp}$ such that $VE = 90\%$, 0.5% of initially infected, 10% of initially immune individuals, and simulations length is set to 1 year.

country showing the highest fraction of 65+ people (24%), while the developing economies show the lowest fraction (Egypt 5.3% and Peru 8.7%). Indeed, low/middle-income countries tend to have a younger population with respect to the high-income ones. Second, we present also the contacts intensity of different age groups. This is defined as the total number of contacts that an individual in a certain age group has, on average, with all the others in a day. We observe a typical decreasing trend, with younger people that tend to have more contacts. In particular, developing economies show a much higher number of daily contacts for individuals aged under 30. This can play an important role in the spreading even in developed economies. According to a recent study the resurgence of the COVID-19 Pandemic in the US after Summer 2020 was mainly sustained by younger people [55]. Finally, we represent a measure of inter-generational mixing. We define it simply as the number of daily contacts that an

individual in the age groups at high mortality risk from COVID-19 (65+) receives from individuals in the age brackets 0–49. We observe that Egypt is the country showing the highest inter-generational mixing, followed closely by Peru.

It is important to notice how heterogeneities in health infrastructures, access to health care, and comorbities are expected across the six countries under examination [54]. These features might induce variations in the IFR. For simplicity however, in the following, we use the same fatality rates for all countries. We leave the exploration of this important aspect to future work.

**Prioritization strategies, rollout speed, and vaccine efficacy.** The proposed model allows to investigate the impact of the behavioural response under different conditions. We consider six populations matching the characteristics of the countries under consideration and exploring the model with the same epidemic initial conditions. The experimental setup considers that each population has already experienced a previous wave of infections and that restrictive measures are in place to mitigate the spreading. Therefore, we set the basic reproductive number $R_0$ = 1.15, 0.5% of initially infected individuals, and 10% of immune individuals. In line with estimates of vaccine efficacy against COVID-19 we set $VE_S$ = 70% and we choose $VE_{Symp}$ such that $VE$ = 90% [57], while we let vary the vaccine rollout speed $r_V$ between 0.1% and 1% to cover the spectrum of real vaccination rollout speeds of the vaccination campaigns across the globe. Peru, for example, administered on average 0.05 daily doses per 100 people in the week commencing on the $8^{th}$ of March, Italy administered 0.30, and Serbia 0.70 [58]. In Fig 1B we show the relative deaths difference for the three vaccine prioritization strategies and the three vaccine rollout speed $r_V$ = 0.1%, 0.5%, 1.0%. The relative deaths difference (*RDD*) is defined as the fraction of deaths averted thanks to the vaccine with respect to a baseline simulation without vaccine and, therefore, no behavioural response. For example, *RDD* = 0.2 indicates that 20% of deaths are averted. Note how this quantity may become negative if the behavioural response causes more deaths than those averted thanks to the vaccine. We refer the reader to the Materials and Methods section for further details. Moreover, we explore a range of behavioural responses running the simulations for a range of values of the parameter $\alpha$. Starting from $\alpha$ = 0 (which implies no behavioural response), we perform simulations with increasing $\alpha$ values (implying stronger reactions), while keeping constant the other behavioural parameter ($\gamma$ = 0.5).

As a first observation, across the different countries and rollout speeds considered, the strategy aiming to curb the severity of the pandemic (i.e., vaccine strategy 1) is indeed the most effective in reducing the number of deaths.

As illustrative and concrete example let us consider the case of Canada. When $\alpha$ = 0 (i.e., no behaviour response) and $r_V$ = 1% with this strategy the fraction of averted deaths is 0.77 with respect to a baseline without vaccine. This fraction reduces to 0.72 with the homogeneous vaccination strategy, and to 0.70 with the strategy prioritizing younger individuals. When $r_V$ = 0.5%, these fractions reduce to, respectively, 0.67, 0.64, 0.62. The ordering of the strategies slightly changes when $r_V$ = 0.1%, in this case we obtain 0.28. 0.35, 0.36. Overall, the picture that emerges is consistent with previous studies on the allocation of COVID-19 vaccines among age groups showing that the strategy minimizing deaths is generally the one targeting the elderly, even though for some combination of the parameters the strategy targeting the younger might be preferable [43, 44].

When the behavioural response to the vaccine rollout is considered (i.e., $\alpha > 0$), we note a consistent decreasing trend of the relative deaths difference for increasing values of $\alpha$. This shows that the behavioural response impacts the unfolding of the epidemic, and that a relaxation of NPIs leads to a smaller fraction of averted deaths. Interestingly, a concerning effect also emerges: in some conditions, as non-compliance becomes larger, the benefit brought by the vaccine is nullified and the number of observed deaths increases with respect to the no-vaccine and no-behaviour change scenario (i.e., the relative deaths difference become negative). This is

solely attributable to the behavioural reaction to the vaccination campaign which in turn is not efficient enough to balance behaviour relaxation. Indeed, this phenomenon is observed in particular when strategies that do not target a reduction in severity are employed, and when the vaccine rollout speed is low. In the case of Serbia, with $r_V$ = 1% and a strategy aimed at reducing severity, the fraction of averted deaths goes from 0.84 when $\alpha$ = 0 to 0.73 when $\alpha$ = 10, with a potential loss of 0.11 attributable to the NPIs relaxation. This effect is more pronounced in the case of the vaccination strategy 2 (or 3): the fraction of averted deaths goes in this case from 0.76 (0.73) when $\alpha$ = 0 to 0.59 (0.50) when $\alpha$ = 10, with a potential loss of 0.17 (0.23) attributable to the NPIs relaxation. Similarly, lower vaccine rollout speed are impacted more significantly by behavioural responses. In the previous example, an immunization campaign with a $r_V$ of 1%, 0.5%, and 0.1% would induce a relative deaths difference, in the case of the vaccination strategy 1, of 0.84, 0.75, 0.40 when $\alpha$ = 0. By setting instead $\alpha$ = 10, these figures would drop to 0.73, 0.44, −0.67 with a loss of 0.11 in the first, of 0.31 in the second, and of 1.07 in the third case.

In the results presented in Fig 1B, we observe that the most disadvantaged countries are the developing economies, Egypt and Peru. At first, this may seem counter-intuitive. Indeed, these countries have a lower fraction of individuals aged over 65. A possible explanation is given by the high activity of young individuals combined with the high inter-generational mixing. Moreover, it is worth stressing that these results are to be intended in relative terms: a relative worst performance in averting deaths does not necessarily imply a worst absolute performance. In other words, the relative impact of behavioral responses might be stronger in those countries, but their age pyramid might induce still a smaller number of deaths in absolute terms. On this point, it is important to remember that we are not considering possible modulations of the IFRs due to the heterogeneity in health care infrastructure, prevalence of co-morbidities and access to health care. In the S1 Text we provide an example highlighting this point by comparing Italy and Egypt. We observe also that the difference between countries is more profound for the vaccination strategy aimed at reducing transmission (i.e., strategy 3). Furthermore, behavioural relaxation widens the distance between the countries. With $r_V$ = 0.1% and vaccination strategy 3, the gap between the relative deaths difference for Italy and Egypt is 0.24 when $\alpha$ = 0. This figure increases to 0.87 when $\alpha$ = 10.

In the S1 Text we repeat the analysis considering the fraction of averted infections instead of deaths. We find that the ranking of allocation strategies is inverted. Indeed, in line with previous findings [43], when considering averted infections the most efficient strategy is the one targeting the younger population first, followed by the homogeneous, and the strategy aimed at curbing severity. In the S1 Text, we also repeat the results presented in Fig 1B exploring different vaccine efficacy $VE$ (50%, 70%, 90%) rather than rollout speed. Similarly to the case presented in the main text, also in these additional analyses we find that early behavioural relaxation reduces the fraction of averted infections, influences heterogeneously the prioritization strategies, and impacts more significantly lower vaccine efficacy or slower rollout.

To further explore how to mitigate the impact of behavioural relaxation once that the vaccination campaign started, in Fig 2 we systematically explored the interplay between vaccine efficacy $VE$ and vaccination rollout speed $r_V$. The black dashed lines highlight the combinations of $VE$ and $r_V$ that achieve a 30% drop in observed deaths (taken as reference value), in the absence of a behavioural response (i.e., $\alpha$ = 0). In most countries, it can be achieved with $r_V$ smaller than 0.2%. On the contrary, when even a mild behavioural response is active (red dash-dotted lines, $\alpha$ = 1), the rollout speed has to increase greatly when vaccine efficacy diminishes: for the case of Italy, a 30% drop in deaths would be achieved with a $r_V$ of 0.4% when the vaccine efficacy is 90%, but in the case of $VE$ = 60% the rollout speed has to grow up to 0.8% to achieve the same result.

**Epidemic trajectories and impact of behavioural relaxation.**   As a way to ground the model on more realistic epidemic scenarios, we calibrate it to the real epidemic trajectories of

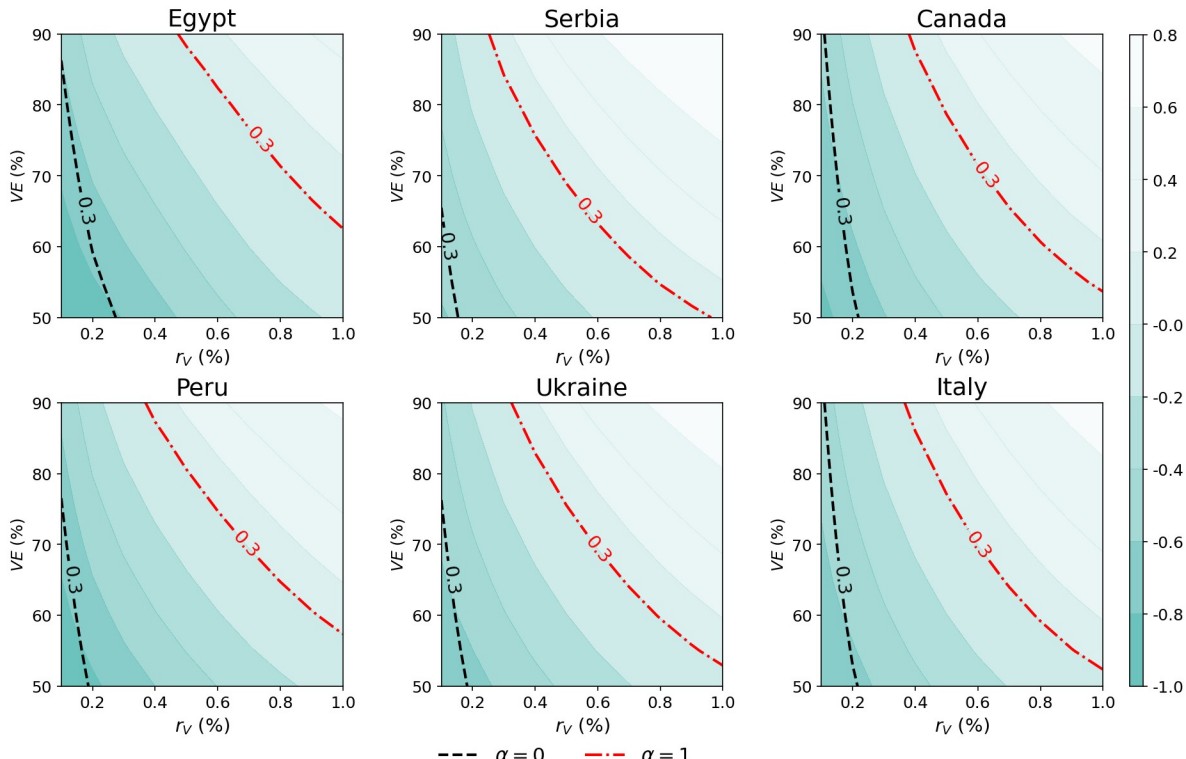

**Fig 2. Interplay between vaccine efficacy and rollout speed.** Contourplot of the relative deaths difference in the scenario with a mild behavioural response ($\alpha = 1$) for different combinations of $VE$ and $r_V$. We let $VE_S$ vary between 30% and 70% and we choose different $VE_{Symp}$ such that the overall efficacy $VE$ vary between 50% and 90%. A 30% reduction of deaths is highlighted with a red dashdotted line. The black dashed line highlight the 30% death drop achieved with a vaccination campaign without the behavioural component ($\alpha = 0$). Vaccination Strategy 1 is considered, parameters used are $\gamma = 0.5$, $R_0 = 1.15$, $r = 1.3$, 0.5% of initially infected, 10% of initially immune individuals, and simulations length is set to 1 year.

the six countries considered. The calibration is performed via an Approximate Bayesian Computation technique (ABC) [59] on weekly deaths for the period 2020/09/01–2020/12/31. In doing so, we estimate some key epidemiological parameters matching the context of each country, thus defining realistic initial conditions for the model. To capture the effects of non-pharmaceutical interventions on the contacts between people we consider data from the Google Mobility Report [60] and the Oxford Coronavirus Government Response Tracker [61] until week 11 of 2021. After the calibration step, the model evolves from 2021/01/01 up to 2021/06/01 with the vaccination rollout for each of the countries. Note that in some of these countries, the vaccination campaigns officially started in the second half of December, though mostly symbolically.

In Fig 3A we show key indicators resulting from the calibration. In particular, for the six countries we report the boxplot for the calibrated infection parameter $\beta$, the projected number of symptomatic infections per 100, 000, and the fraction of recovered individuals on the 2021/01/01, which is the start of the vaccination campaign in our simulations. We acknowledge a significantly high acquired immunity in the case of Peru, where the estimated attack rate as of $1^{st}$ January, 2021, is around 38%. Nonetheless, this figure is line with other available estimates [58]. We observe how the calibration allows for an heterogeneous representation of the epidemiological conditions of the different countries. In Fig 3A we also show the effects of restrictive measures on contacts. As a proxy, we consider the ratio between the leading eigenvalue of the contacts matrix (considering the restrictions) and the leading eigenvalue of the baseline contacts matrix (without restrictions). The leading eigenvalue of the contacts matrix influences the

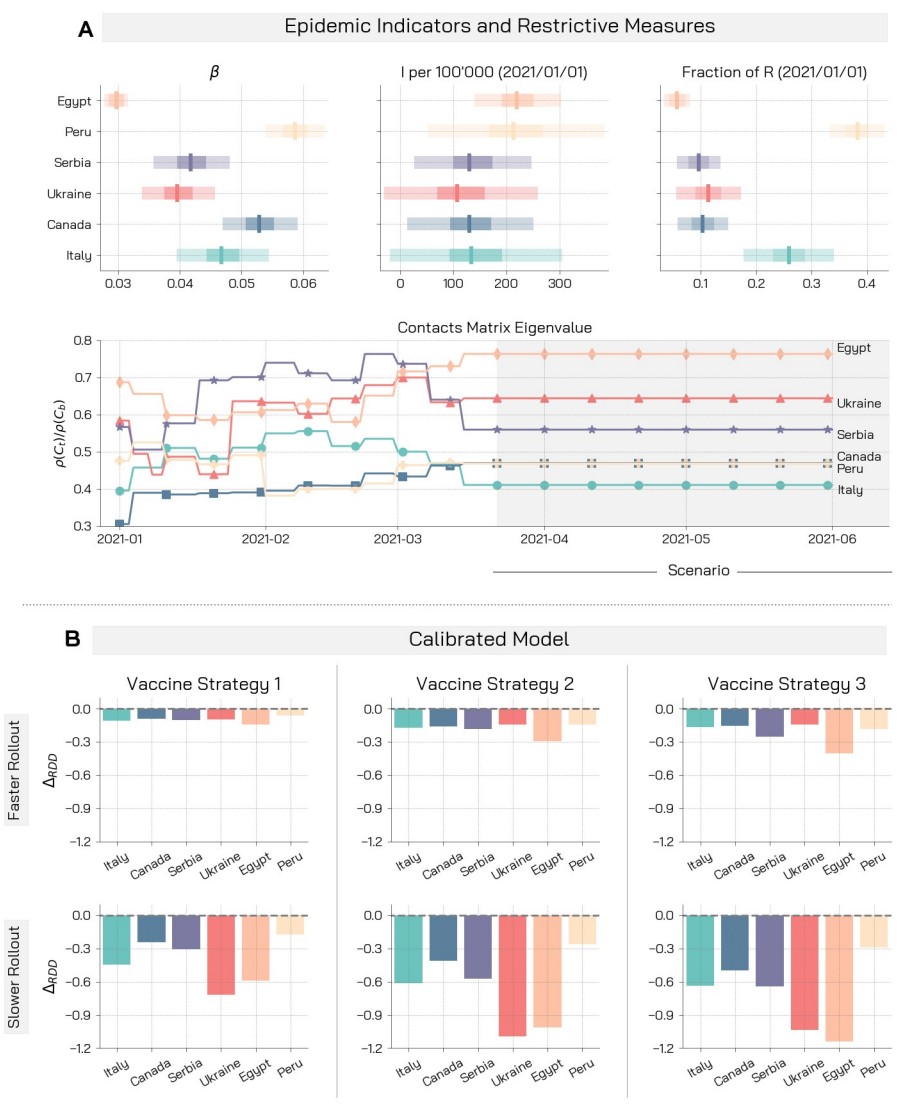

**Fig 3. Giving up NPIs during rollout may nullify the benefits brought by the vaccine.** A) We display for the different countries the boxplot of the calibrated infection parameter $\beta$, the projected number of symptomatic infectious cases per 100'000 and the fraction of recovered at the 2021/01/01, start of the vaccination campaign in our simulation. We also report the ratio between the leading eigenvalue of the contacts matrix considering restrictions and of the baseline contacts matrix with no restrictions. B) We display the median relative deaths difference for the calibrated model in the different countries. We consider the three vaccination strategies and two possible rollout speed: $r_V = 1\%$ (faster rollout), and $r_V = 0.25\%$ (slower rollout). We run the model over the period 2021/01/01–2021/06/01. Other parameters are $\gamma = 0.5$, $\alpha = 10^2$, $r = 1.3$, $VE_S = 70\%$ ($VE = 90\%$).

reproductive number of the disease (see Materials and methods), and more broadly it measures the intensity of contacts among people. By normalizing with respect to the baseline contacts matrix (with no restrictions) we can grasp the strictness of the measures in place. For example, a ratio of 0.3 would imply, in our simulations, a 70% reduction of the reproductive number with respect to the baseline without restrictions. The effect of restrictive measures on contacts varies over time up to week 11, 2021. In the case of Italy, for example, we observe the partial ease of the measures during January and February (i.e., $\rho(C_t)/\rho(C_b)$ increases) followed by the tightening of measures in March aimed at curbing the third wave of infections. Afterwards, we keep the mixing levels as those observed for week 11 (more details in the Materials and methods

section). This is a conservative assumption as we can imagine that the seasonality and impact of the vaccination campaign might induce a relaxation of NPIs as the number of cases and deaths, hopefully, will go down. In this scenario, the countries with the strictest measures in place after week 11, are Peru and Italy, and Canada, while Egypt and Ukraine show the most permissive measures. In the S1 Text, we repeat the analysis using real data to compute contacts reduction parameters for the whole period. The results are in line with the observations presented below.

In Fig 3B we study, for the calibrated model in the six countries, the difference between the fraction of averted deaths with respect to a baseline without vaccine in the case of a strong behavioural reaction to vaccine deployment (i.e., $\alpha = 10^2$) and in the case of no relaxation of NPIs (i.e., $\alpha = 0$). This quantity, which we indicate with $\Delta_{RDD}$, represents the additional fraction of deaths that occur due to lower compliance to COVID-safe behaviours in our simulations (see Materials and methods for more details). We study two vaccination rates. A fast deployment that manages to vaccinate $r_V = 1\%$ of the population daily (aligned with the preparedness plan for Influenza pandemic [62] and vaccination rates achieved by countries like Israel or Chile), and a slower rollout $r_V = 0.25\%$. Worryingly, in the different countries, even in a scenario with several restrictions and a successful vaccination campaign in place, it is possible to witness a significant increase in deaths exclusively due to a relaxation of COVID-safe behaviours. As a result of the calibration step, the countries under investigation face different epidemiological conditions (e.g., effective reproductive number, immunity from previous waves, initial number of infected, estimated effects of restrictions). These translate into phenomenological differences with respect to the results of the previous analyses. In particular, Ukraine and Egypt are significantly impacted by behavioural relaxation in the case of the slower rollout $r_V = 0.25\%$, while Peru, differently from the previous analysis, is less impacted. From Fig 3A we see that in the first two countries the estimated effect of restrictions on contacts (especially in the last weeks) is smaller with respect to the others. On the other hand, Peru features tougher NPIs, together with Italy and Canada, which are also less impacted by behaviour in Fig 3B. This underlines how, besides the characteristics of the vaccination campaign, also the epidemiological conditions and the measures in place are important factors influencing the behaviour-vaccine interplay. In the case of the faster rollout $r_V = 1\%$, the obtained values of $\Delta_{RDD}$ are smaller and the differences between countries are less pronounced. A possible explanation is that when a fast rollout is employed the influence of epidemiological conditions and restrictions become less important because of the efficiency of vaccine administration. Furthermore, we observe that, for both $r_V$, the vaccination strategy targeting a reduction of severity (strategy 1) performs generally better in case of NPIs relaxation with respect to the other strategies. In the S1 Text we investigate more in detail the comparison between vaccination strategies in terms of robustness to behaviour relaxation considering both averted deaths and infections. We find that, when considering deaths, a vaccine allocation aimed at reducing disease severity (i.e., strategy 1) is always preferable. Instead, vaccination strategy 2 and 3 perform better when the number of avoided infections is used as evaluation metric. Information about vaccination strategy robustness against behavioural changes might be used to tune and design resilient rollout campaigns. Policy-makers might also consider to optimize vaccination strategies with respect to a combination of multiple metrics. In this direction, our analysis provide additional insights, and suggests that behavioural changes might play an important role possibly modifying the impact of vaccine prioritization strategies.

## Discussion

For almost a year, in the midst of a global pandemic, policymakers struggled to implement sustainable restrictions to slow SARS-CoV-2 spreading. Every non-pharmaceutical intervention

was aimed at slowing (in few countries stopping) the disease progression buying time for the development, test, production and distribution of vaccines that might ultimately protect the population. With an impressive scientific endeavor, several vaccines have been developed and an early distribution campaign was rolled out by the last days of December 2020. Besides the potential threats emerging from new virus strains [63], the current vaccination campaign represents the beginning of a new normal and a gigantic step towards complete virus suppression. However, we demonstrated that if the growth in vaccination uptake would lead to overconfident conducts inducing relaxation of COVID-safe behaviours, additional avoidable deaths will occur. We extended the literature proposing a mechanistic compartmental model able to simulate the unfolding of COVID-19, the vaccination dynamics and the compliance/non-compliance transition modulated by different behavioural mechanisms. Performing in-silico simulations allowed us to explore, from a theoretical standpoint, the interplay among different vaccination strategies, rollout speeds, vaccine efficacy, and behavioural responses. We found that behaviour impacts non-linearly vaccine effectiveness. Indeed, early NPIs relaxation affects more significantly slower rollouts, lower vaccine efficacy, and allocation strategies that target reduction in transmission rather in severity. We included in our analysis six different countries (Egypt, Peru, Serbia, Ukraine, Canada, and Italy), representatives of various points of the spectrum of world economies. This allowed us to observe the effects of behaviour-vaccine relationships for various population pyramids and mixing patterns. We observed that the developing economies, characterized by a younger population, but higher contacts activity and inter-generational mixing, are generally more affected by behaviour change with respect to developed economies. Then, as a way to ground the model on more realistic epidemiological conditions, we calibrated it using real epidemic and mobility data for the six countries considered and simulated the unfolding of the first months of the vaccination campaign. In such realistic scenario, we observed that even with restrictive measures in place and a successful vaccination campaign, it is possible to witness to non-negligible increases in deaths attributable to an early relaxation of COVID-safe behaviours. The calibration step allowed us to highlight that also the epidemiological conditions related to the country-specific unfolding of the disease are an important factor influencing the interplay behaviour-vaccine.

We acknowledge some limitations in the present study. First, we considered the vaccines fully working immediately after the first dose and we neglected that vaccination campaigns are using a portfolio of vaccines rather than a single one. We have also studied three simple vaccination strategies that neglect the complexities of an unprecedented mass vaccination. As result, both the vaccination priorities, vaccine effects and vaccination rates are an approximation of reality. In the S1 Text, we studied a data-driven vaccination rollout for Italy, where vaccines are distributed to the various age brackets following the real daily administration data. The results we obtained are qualitatively similar to those presented in the main text. Second, while the model calibration suggests that our approach can nicely capture national trends, the model is not meant to provide accurate forecasts of the local unfolding of the disease, but rather to test what-if scenarios in a comparative fashion. We have considered a simple age-structure compartmental model that does not capture spatio-temporal heterogeneity both in terms of spreading and of NPIs implementation which have instead been observed in the countries under investigation. Third, our model does not include new, more transmissible, variants of concern and assumes the same IFR across the six countries. Finally, we propose and model two potential mechanisms leading to behavioural changes, but data are not available to perform a quantitative validation of the behavioural components of the model.

Implementation of individual protective behaviours and adherence to NPIs have been vital in order to reduce the transmission of SARS-CoV-2 leading to substantial population-level effects [5, 64–75]. Behavioural science can provide valuable insights for managing policies,

incentives, communication strategies and can help coordinate efforts to control threats and evaluate such interventions [76]. As during the first waves of COVID-19, when NPIs were the only available mitigation measures [77], the results of our paper call for adequate strategies to keep high the attention and compliance towards individual COVID-safe behaviours, such as mask-wearing, social distancing, and avoidance of large gatherings now that vaccines are finally available. Communication strategies and policies should keep targeting such non-pharmaceutical intervention to avoid frustrating the immense effort of the vaccination campaigns.

## Materials and methods

### Epidemic model

**Model definition.** We propose an age-stratified compartmental model that incorporates both the vaccination process and the behaviour dynamics possibly linked to it. See Fig 4 for a schematic representation. Individuals are divided into 16 five-year age groups (last group is 75+). The virus transmission is modeled using the following approach. Susceptible individuals ($S$ compartment) in contact with infectious compartments become latent ($L$ compartment). After the latent period ($\epsilon^{-1}$), $L$ individuals enter the pre-symptomatic stage of the infection ($P$). Then, they can transition either in the asymptomatic ($A$) or the symptomatic stage ($I$) at rate $\omega$ (the length of time spent in the $L$ and $P$ compartments is the incubation period). The probability of being asymptomatic is $f$. After the infectious period ($\mu^{-1}$), both $I$ and $A$ individuals enter the Recovered compartment ($R_I$, $R_A$). Alternatively, $I$ individuals can also die and transit to the $D$ compartment. Note how we include a delay of $\Delta$ days from the time individuals enter the

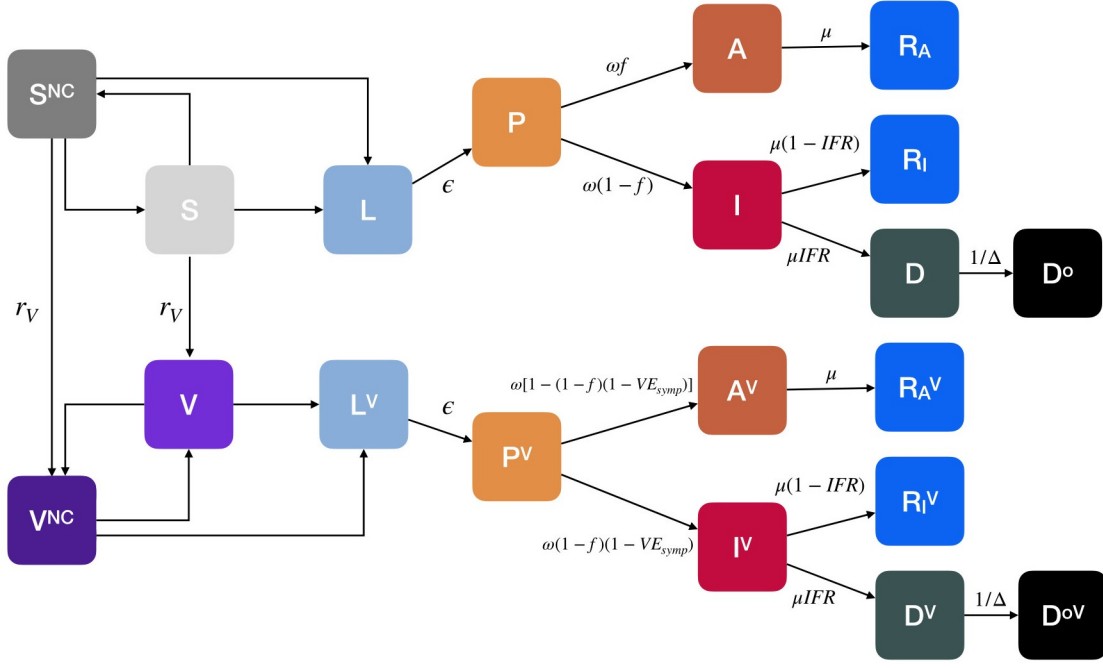

**Fig 4. Compartmental model.** We consider an extension of the classic SLIR model adding compartments for presymptomatic ($P$) and asymptomatic ($A$). We also design compartments for vaccinated ($V$), dead ($D$ describes individuals that will die with a delay of $\Delta^{-1}$ entering the compartment $D^o$), susceptible ($S^{NC}$) and vaccinated ($V^{NC}$) individuals that do not comply with COVID-safe behaviours. The vaccine offers a protection $VE_S$ against infection and $VE_{Symp}$ against symptoms that can lead to severe outcomes such as death. The transmission rate for susceptible is $\beta$ and for susceptible non-compliant $r\beta$ ($r > 1$). The parameter $\alpha$ regulates the transition from compliant to non-compliant behaviours, while $\gamma$ regulates the opposite flow. Arrows describe the transitions between compartments. For simplicity of visualization, we do not display the compartments mediating the different transitions. For example, the transition $S \rightarrow L$ is mediated by the infectious compartments ($P$, $A$, $I$, $P^V$, $I^V$, $A^V$). The compartmentalization is then extended to account for empirical age-structure and contact matrices.

compartment $D$ and die ($D^o$ compartment, the superscript stands for "observed"). The infectious compartments are $P$, $A$, $I$. The transmission rate is $\beta$ and the force of infection is dependent on the age-stratified contact matrix $\mathbf{C} \in \mathbb{R}^{K \times K}$, whose element $C_{ij}$ represents the average number of contacts that an individual in age group $i$ make with individuals in $j$ per day. The matrix $\mathbf{C}$ has four location-specific contributions: contacts at home, workplace, school, and other locations. We adopt country-specific contacts matrices provided in Ref. [78]. We assume that $P$ and $A$ have lower infectiousness with respect to symptomatic $I$ ($\beta\chi$, with $\chi < 1$). Similar approaches have been used in several modeling studies in the context of the current pandemic [70, 79].

On top of the disease dynamics, we model both the vaccination process and the behavioural change that is possibly coupled to the vaccination. More in detail, after the start of the vaccination campaign at $t_V$, at each time step, a fraction of the susceptible population receive a vaccine and transit to compartment $V$. We introduce the rollout speed $r_V$ as the number of daily available doses expressed as a percentage of the total population. We adjust the number of doses available per day considering the fraction of susceptibles among all those that can receive the vaccine ($S$, $L$, $P$, $A$, and $R_A$) and we assume that the remaining doses are wasted. We consider a "leaky" vaccine that reduces susceptibility with a certain efficacy $VE_S$ and the probability of developing symptoms by $VE_{Symp}$. In other words, the infection rate for $V$ individuals is $\beta(1 - VE_S)$ and the probability of entering the infectious symptomatic compartment $I^V$ from $P^V$ is $(1 - f)(1 - VE_{Symp})$. Similar approaches have been used in previous works in the context of mathematical modeling of COVID-19 immunization campaigns [44, 80]. As mentioned above, we consider three vaccination strategies: one in which the vaccine is given in decreasing order of age (vaccination strategy 1 aimed at reducing the severity), one in which it is given homogeneously to the population (vaccination strategy 2), and one in which it is first given to individuals in age brackets 20–49 and then to the rest of the population (vaccination strategy 3 aimed at reducing the transmission of the virus) [10]. In parallel, we imagine a behavioural dynamics triggered by the presence of the vaccine. Indeed, susceptible individuals (both vaccinated and not) may start adopting less safe behaviours because reassured by the presence of an effective vaccine. This is encoded in the model with a transition from the compartment $S$ ($V$) to a new compartment $S^{NC}$ ($V^{NC}$)—NC stands for non-compliant—of individuals that protect less themselves and as a result get infected at a higher rate. The parameter $r > 1$ captures the increased risk of contagion for $NC$ individuals. It modulates the contribution to the force of infection from the $S^{NC}$ and $V^{NC}$ compartments. In the simulations presented in the main text we used $r = 1.3$ imagining a scenario where the relaxation of COVID safe behaviors increases the transmission rate of 30%. This choice is informed by the estimated effects of NPIs such as face masks and social distancing on COVID-19 spreading, as reported in the literature [4, 20, 81, 82]. Nonetheless, in the S1 Text we perform a sensitivity analysis considering also $r = 1.1$, 1.5. We propose two mechanisms to model the behavioural transition. In the first mechanism, the transition towards non-compliance happens at a rate $\alpha$ and it is catalyzed by the cumulative fraction of individuals that received a vaccine ($v_t$, including both compliant and non-compliant). The opposite transition from $S^{NC}$ ($V^{NC}$) to $S$ ($V$) happens at rate $\gamma$ and is catalyzed by the number of deaths per 100, 000 observed in the previous time step ($d^o_{t-1}$, including both compliant and non-compliant). Indeed, an increase in deaths is frequently used—especially by media—as an indicator of the severity of the current epidemiological situation. Existing literature suggests that risk perception (in the form of number of infected individuals or deaths) and communication of such risk significantly affect adherence to personal mitigation strategies such as social distancing and wearing face masks [5, 83]. Expressing the number of deaths in proportion to the population allows us to compare countries of different size. Overall, this approach aims to depict the adaptive nature of human behaviour where individual choices are

influenced by vaccination and pandemic progression [84]. In the second mechanism, $S$ ($V$) individuals transit to the non-compliant compartment $S^{NC}$ ($V^{NC}$) at a constant rate $\alpha$. We also account for the possibility of non-compliant going back to safer behaviours again at a constant rate $\gamma$. To simplify the narrative, we present the results considering this second mechanism only in the S1 Text. The overall picture discussed above does not change significantly. Note that in order to avoid issues with transition probabilities large than one we model the rates as $\lambda_{X \to X^{NC}} = 1 - \exp^{-g(\alpha)}$, $\lambda_{X^{NC} \to X} = 1 - \exp^{-h(\gamma)}$ with $X = [S, V]$ and where the specific expression of the exponent depend on the two mechanisms described above: $g(\alpha) = \alpha v_t$, $h(\gamma) = \gamma d_{t-1}^o$ in the first one, and $g(\alpha) = \alpha$, $h(\gamma) = \gamma$ in the second mechanism. Note how for small values of $g(\alpha), h(\gamma)$ the rate converges to the usual mass-action law. The model just described can be written down as the following system of differential equations for individuals in age group $k$:

$$\frac{dS_k}{dt} = -\lambda_k S_k - (1 - e^{-g(\alpha)})S_k + (1 - e^{-h(\gamma)})S_k^{NC}$$

$$\frac{dS_k^{NC}}{dt} = -r\lambda_k S_k^{NC} + (1 - e^{-g(\alpha)})S_k - (1 - e^{-h(\gamma)})S_k^{NC}$$

$$\frac{dL_k}{dt} = +\lambda_k S_k + r\lambda_k S_k^{NC} - \epsilon L_k$$

$$\frac{dP_k}{dt} = \epsilon L_k - \omega P_k$$

$$\frac{dI_k}{dt} = \omega(1 - f)P_k - \mu I_k$$

$$\frac{dA_k}{dt} = \omega f P_k - \mu A_k$$

$$\frac{dR_k}{dt} = \mu(1 - IFR_k)I_k + \mu A_k$$

$$\frac{dD_k}{dt} = \mu IFR_k I_k - \Delta^{-1} D_k$$

$$\frac{dD_k^o}{dt} = \Delta^{-1} D_k$$

$$\frac{dV_k}{dt} = -(1 - VE_S)\lambda_k V_k - (1 - e^{-g(\alpha)})V_k + (1 - e^{-h(\gamma)})V_k^{NC}$$  (1)

$$\frac{dV_k^{NC}}{dt} = -r(1 - VE_S)\lambda_k V_k^{NC} + (1 - e^{-g(\alpha)})V_k - (1 - e^{-h(\gamma)})V_k^{NC}$$

$$\frac{dL_k^V}{dt} = +(1 - VE)\lambda_k V_k + r(1 - VE)\lambda_k V_k^{NC} - \epsilon L_k^V$$

$$\frac{dP_k^V}{dt} = \epsilon L_k^V - \omega P_k^V$$

$$\frac{dI_k^V}{dt} = \omega(1 - f)(1 - VE_{Symp})P_k^V - \mu I_k^V$$

$$\frac{dA_k^V}{dt} = \omega(1 - (1 - f)(1 - VE_{Symp}))P_k^V - \mu A_k^V$$

$$\frac{dR_k^V}{dt} = \mu(1 - IFR_k)I_k^V + \mu A_k^V$$

$$\frac{dD_k^V}{dt} = \mu IFR_k I_k^V - \Delta^{-1} D_k^V$$

$$\frac{dD_k^{Vo}}{dt} = \Delta^{-1} D_k^V$$

Where $\lambda_k = \beta \sum_{k'=1}^{K} C_{kk'} \frac{I_{k'} + I_{k'}^V + \chi(P_{k'} + A_{k'} + P_{k'}^V + A_{k'}^V)}{N_{k'}}$ is the force of infection for age group $k$.

The basic reproduction number is $R_0 = \rho(\tilde{C})[\frac{\beta\chi}{\omega} + \frac{\beta(1-f)}{\mu} + \frac{\beta\chi f}{\mu}]$, where $\tilde{C}$ is the contacts matrix weighted by the relative population in different age groups. In the S1 Text we provide details on its derivation. We adopt the model integrating numerically the equations, thus it is deterministic. However, it is worth stressing that when we calibrate the model to the real epidemic trajectory in the six countries, we use a probabilistic framework through an Approximate Bayesian Computation technique. Said differently, the calibrated parameters are characterized by posterior probability distributions rather than exact values. For this reason, the results presented in Fig 3 (i.e., the median of model's projections) are to be intended as an ensemble of multiple trajectories generated sampling from the posterior distribution of the fitted parameters. The model is implemented in the programming language python with the use of the libraries *scipy* [85], *numpy* [86], and *numba* [87]. The visualization of the results is realized with the library *matplotlib* [88].

In the Results section, we kept for simplicity $\gamma = 0.5$ and we let vary $\alpha$. The choice of $\gamma$ is informed by the maximum number of deaths observed in the countries of focus. In Italy, for example, the maximum number of deaths reported on a single day is around 1,000. Therefore, this value of $\gamma$ is such that, in a similar situation, non-compliant individuals would likely return to COVID-safe behaviours. As described in the previous paragraph, in the dynamic-rate behavioural mechanism we model the transition rate as $\lambda_{X^{NC}\to X} = 1 - \exp^{-\gamma d^o_{t-1}}$ ($X = [S, V]$), where $d^o_{t-1}$ is the number of deaths per 100,000 individuals observed at time $t-1$. Hence at the peak of deaths the transition rate towards compliance is $\lambda_{X^{NC}\to X} \sim 0.6$. The sensitivity to this choice is discussed in the S1 Text, where we also report additional analyses to better understand the effects of the behavioural parameters. More in detail, we plot the rates $\lambda_{X\to X^{NC}}$ and $\lambda_{X^{NC}\to X}$ together with the actual number of *NC* individuals in time for several values of $\alpha$, $\gamma$, $\nu_t$, $d^o_{t-1}$ and different epidemiological conditions.

In Table 1 we report a list of the model's parameters together with their values used in the simulations and the related sources. We also indicate which values are optimized during the calibration step (described in the next sections).

**Relative deaths difference.** In the previous analyses, we compared the vaccination strategies in the six countries using different evaluation metrics. The relative deaths difference (*RDD*) represents the fraction of deaths that are avoided in a simulation with vaccines (and behavioural response) with respect to a baseline simulation without vaccines (and no behavioural response). We compute it as:

$$RDD(\alpha) = \frac{deaths^{novaccine} - deaths^{vaccine}(\alpha)}{deaths^{novaccine}}$$

**Table 1. Model parameters.**

| Parameter | Symbol | Value | Source |
|---|---|---|---|
| Transmissibility | $\beta$ | calibrated | calibration range informed by Ref. [89] |
| Latent period | $\epsilon$ | 3.7 days | [90, 91] |
| Presymptomatic period | $\omega$ | 1.5 days | [90, 91] |
| Fraction of asymptomatic | $f$ | 0.35 | [92, 93] |
| Reduced infectiousness of P and A | $\chi$ | 0.55 | [94] |
| Infectious period | $\mu$ | 2.5 days | [95, 96] |
| Infection fatality rate | IFR | age-stratified | [56] |
| Days spent in $D$ before transitioning to $D^o$ | $\Delta$ | calibrated | calibration range informed by Ref. [97] |

Where we made explicit the dependence from $\alpha$. By definition, a value of $RDD = 0.2$ indicates that 20% less deaths are observed in the simulation with vaccines administered. Negative values of $RDD$ capture instead scenarios where the behavioral response induces more deaths that in the case without vaccines. An analogous quantity can be easily computed for infections. In Fig 3B we have also considered the difference between the fraction of averted deaths with respect to a baseline without vaccine in the case of behavioural reaction (i.e., $\alpha > 0$) and in the case of no relaxation of NPIs (i.e., $\alpha = 0$). We indicated this quantity with $\Delta_{RDD}(\alpha)$, and we compute it as follows:

$$\Delta_{RDD}(\alpha) = RDD(\alpha) - RDD(0)$$

With respect to the simple $RDD$, the $\Delta_{RDD}$ provides a slightly different piece of information. Indeed, by subtracting the $RDD$ obtained with $\alpha = 0$ we isolate only the effects of behaviour. As a clarifying example, $\Delta_{RDD}(\alpha) \sim 0$ refers to a negligible effect of behaviour on the fraction of averted deaths. Instead, $\Delta_{RDD}(\alpha) < 0$, indicates that due to behaviour relaxation more deaths are occurring. Also in the case of $\Delta_{RDD}(\alpha)$, an analogous for infections can be easily derived.

## Vaccination strategies

We consider three vaccination strategies. In the first one, the vaccine is given in decreasing order of age. Since the IFR of COVID-19 strongly correlates with age, many countries worldwide are adopting similar strategies prioritizing the vaccination among the elderly. Previous modeling works in the context of COVID-19 showed that this is the preferable strategy when considering the number of averted deaths [10, 43, 44]. In practice, this means that in our simulations we start giving the vaccine to the 75+ age bracket and we proceed in decreasing order only when everyone in this group is vaccinated. In the second strategy the vaccine is given homogeneously to the population respecting the age distribution. This means that, of the $X$ vaccines available at step $t$, the fraction $N_k/N$ is given to age group $k$ (where $N_k$ is the number of individuals in age bracket $k$, and $N$ is the total number of individuals). In the third strategy, we prioritize age groups 20–49. Indeed, individuals at high mortality risk from COVID-19 disease may be protected indirectly by vaccinating age brackets that sustain the transmission [98, 99]. In practice, in our simulations we start giving the vaccine homogeneously to the 20–49 age brackets, and when they are all vaccinated we give it homogeneously to all other groups. In the previous two strategies, if, on a given time step, the number of people remained in the age group which is currently being vaccinated is smaller than the total number of vaccines available, the exceeding part is given to the next age group in line. This implies that the number of vaccines given in different time steps is always constant. Since doses may be administered to non-susceptible, on each day we compute the fraction of $S$ individuals among all those who may have received a vaccine (latent, presymptomatic, infectious asymptomatic, and recovered asymptomatic individuals) and we adjust accordingly the number of doses available. A similar approach has been used in Refs. [44, 80]. Though a simplification of reality this approach allows to avoid making assumptions about the possible modulations of vaccine efficacy for individuals vaccinated when not susceptible. In the S1 Text, we repeat part of the analyses considering that some individuals may decide not to get vaccinated. We find that, despite shifts due to vaccine hesitancy, the main findings remain unchanged.

## Model calibration

In this section we describe the methods used to calibrate the model to the real epidemic trajectories in the countries considered. We use an Approximate Bayesian Computation technique

to calibrate the model on weekly deaths during the period 2020/09/01–2020/12/31. We account for government-mandated restrictions and their effect on the contacts between individuals modifying the contacts matrices using data from the Google Mobility Report [60] and the Oxford Coronavirus Government Response Tracker [61].

**Approximate Bayesian computation.** We calibrate the model for each country using the Approximate Bayesian Computation (ABC) rejection method [59, 100]. At each step of the rejection algorithm, a set of parameters $\theta$ is sampled from a prior distribution and an instance of the model is generated using $\theta$. Then, an output quantity $E'$ of the model is compared to the corresponding real quantity $E$ using a distance measure $s(E', E)$. If this distance is greater (smaller) than a predefined tolerance $\psi$, then the sampled set of parameters is discarded (retained). After accepting $N$ sets, the iteration stops and the distribution of accepted parameters is an approximation of the real posterior distribution $P(\theta, E)$. The free parameters of our model and the related prior distributions are:

- The transmission rate $\beta$. We explore uniformly values of $\beta$ such that the related $R_0$ is between 0.8 and 2.2. The basic reproduction number of SARS-CoV-2 is higher [101], but we consider lower values since our calibration starts on 2020/09/01 when restrictions were in place to mitigate the spreading.

- The delay in deaths $\Delta \sim [14, 25]$. Indeed, for COVID-19 the average time between symptoms onset and death is about 2 weeks [97] and we also account for possible additional delays in death reporting.

- The initial number of infected individuals. We explore uniformly values between 0.5 and 15 times the number of reported cases in the week before the start of the simulation. We then assign these individuals to the infected compartments ($L$, $P$, $A$, $I$) proportionally to the time spent there by individuals ($\epsilon^{-1}$ for $L$, $\omega^{-1}$ for $P$, and $\mu^{-1}$ for $I$, $A$), and we split between $I$ and $A$ individuals considering the fraction of asymptomatic $f$.

The model is calibrated on the period 2020/09/01–2020/12/31 using the weighted mean absolute percentage error on weekly deaths as an error metric with a tolerance $\psi = 0.3$ ($\psi = 0.4$ for Egypt for convergence issues) and 5, 000 accepted parameters set.

**Model initialization.** The number of individuals in different age groups is initialized considering the 2019 United Nation World Population Prospects [102]. We consider 16 age brackets of five-years, except for the last one that includes 75+ individuals. As specified above, the initial number of infected individuals is calibrated considering the total number of confirmed cases in the week before the start of the simulation according to the data issued from the European Centre for Disease Prevention and Control [103]. To initialize the number of non-susceptible individuals (placed in the $R$ compartment) we compute the average of several publicly available projections of total COVID-19 infections up to 2020/09/01 (i.e., start of the simulation) from different modeling approaches [58]. Both the initial number of infected and of non-susceptibles is assigned homogeneously across age groups. Other parameters used are $\epsilon^{-1} = 3.7$ days, $\mu^{-1} = 2.5$ days, $\omega^{-1} = 1.5$ days $\chi = 0.55$, $f = 0.35$ in line with current estimates of COVID-19 infection dynamics parameters [90–92, 95, 96]. We use the age-stratified Infection Fatality Rate (IFR) from Ref. [56].

**Modeling the effects of NPIs on contact matrices.** In our model, we incorporate the implementation of top-down NPIs by changing the contacts patterns defined by the contacts matrix **C**. As mentioned, we use the country-specific contacts matrices provided in Ref. [78]. These are made by four contributions: contacts that happen at school, workplace, home, and other locations. In a baseline scenario, the overall contacts matrix **C** is simply the sum of these four contributions. Following Ref. [104], we implement the reductions in contacts, due to the

restrictions, multiplying the single contribution by a reduction factor $\eta_i(t)$. Thus, in general the overall contacts matrix at time $t$ become:

$$C(t) = home + \eta_w(t) \cdot work + \eta_s(t) \cdot school + \eta_{ol}(t) \cdot other\ locations \qquad (2)$$

For simplicity, we assume no changes to contacts at home, though lockdowns tend to increase them [64]. For the contacts locations *work* and *other locations* we use data from the Google Mobility Report [60]. In detail, we consider the fields `workplaces percent change from baseline` to model reduction in the contacts location *work* and the average of the fields `retail and recreation percent change from baseline` and `transit stations percent change from baseline` for *other locations*. A general entry of the report $p_l(t)$ represents the percentage change on day $t$ of total visitors to a specific location $l$ with respect to a pre-pandemic baseline. From $p_l(t)$ we derive a contacts reduction coefficient as follows: $\eta_l(t) = (1 + p_l(t)/100)^2$. Indeed, the number of potential contacts in a specific location is proportional to the square of the the number of visitors. We model contacts reduction in school considering instead the Oxford Coronavirus Government Response Tracker [61]. More in detail, we use the ordinal index `C1 School closing`. The index ranges from a minimum of 0 (no measures) to a maximum of 3 (require closing all levels). We turn this quantity into the contacts reduction coefficient in school as follows: $\eta_s(t) =$ `(3 − C1 School closing)`$/3$. We use these datasets to inform contacts reductions up to week 11, 2021. After, we assume that contacts remain at the same level of week 11. Note how

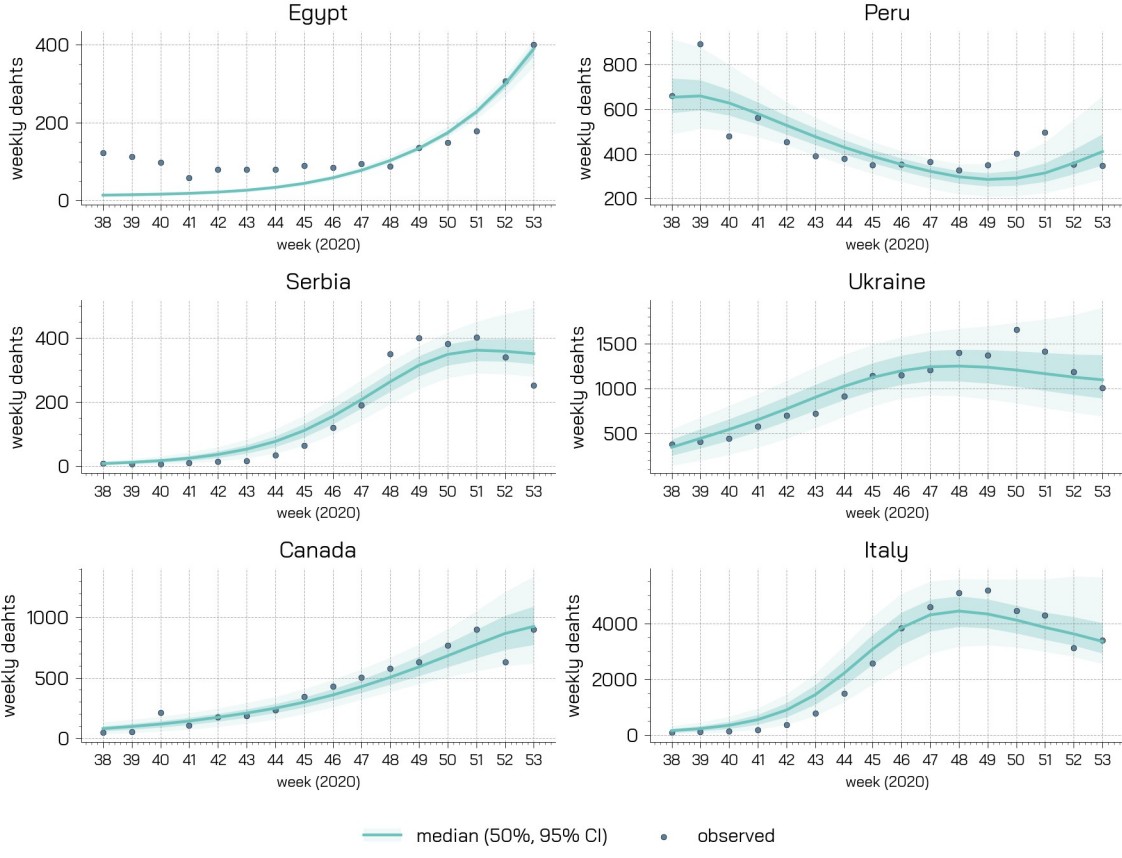

**Fig 5. Calibration results.** For each country we represent the observed and simulated weekly deaths (median, 50% and 95% confidence intervals).

we let vary both the transmission rate $\beta$ and the contact matrix. The former describes the risk of infection given contacts with infectious individuals. This is function of the disease (which is assumed to be the same, we don't consider multiple or emergent new strains possibly more transmissible) and of the protective behaviours such as social distancing and use of face masks. The latter describes variations to the number and types of contacts induced by top-down NPIs as for example remote working, schools closure and lockdowns. By splitting the contributions to the force of infection of transmission rate and contact matrix we are able to take into consideration different behavioural attitudes which, given the same number of contacts, might lead to higher or lower risks of infection. This allows us to consider explicitly both top-down and bottom-up NPIs.

**Calibration results.** In Fig 5, we report the results of the calibration. It is important to stress how our goal is not to develop a predictive model aimed at forecasting the pandemic trajectory. The fit is used to ground the model and to define the epidemic conditions at the start of the vaccination campaign in the six countries. In fact, our aim is to understand the possible interplay between behaviours and vaccine rollout which is also function of the epidemic progression. In the Figure we report the official and simulated weekly number of deaths (median, 50% and 95% confidence interval). Despite its simplicity and approximations, the model is able to reproduce the evolution of the pandemic in the six countries capturing well its progression after the summer. In the S1 Text we also report the posterior distributions for the parameters.

## Supporting information

**S1 Text. Supplementary and sensitivity analysis to the modeling framework.** In this supplementary material we present additional analyses and extensions of our model. We also run extensive sensitivity analysis on the parameters used in the main text.
(PDF)

## Acknowledgments

All authors thank the High Performance Computing facilities at Greenwich University.

## Author Contributions

**Conceptualization:** Nicolò Gozzi, Paolo Bajardi, Nicola Perra.

**Data curation:** Nicolò Gozzi.

**Investigation:** Nicolò Gozzi, Paolo Bajardi, Nicola Perra.

**Methodology:** Nicolò Gozzi, Paolo Bajardi, Nicola Perra.

**Software:** Nicolò Gozzi.

**Supervision:** Paolo Bajardi, Nicola Perra.

**Writing – original draft:** Nicolò Gozzi, Paolo Bajardi, Nicola Perra.

**Writing – review & editing:** Nicolò Gozzi, Paolo Bajardi, Nicola Perra.

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
