## [Decision Letter · Decision Letter 0]

27 May 2021

Dear Dr. Bajardi,

Thank you very much for submitting your manuscript "The importance of non-pharmaceutical interventions during the COVID-19 vaccine rollout" for consideration at PLOS Computational Biology.

As with all papers reviewed by the journal, your manuscript was reviewed by members of the editorial board and by several independent reviewers. In light of the reviews (below this email), we would like to invite the resubmission of a significantly-revised version that takes into account the reviewers' comments. Please make sure that the revision also includes the link to the model code (e.g. as a link to a source-repository). 

We cannot make any decision about publication until we have seen the revised manuscript and your response to the reviewers' comments. Your revised manuscript is also likely to be sent to reviewers for further evaluation.

Sincerely,

Roger Dimitri Kouyos

Associate Editor

PLOS Computational Biology

Virginia Pitzer

Deputy Editor-in-Chief

PLOS Computational Biology

Reviewer's Responses to Questions

**Comments to the Authors:**

Reviewer #1: Summary

The paper by Gozzi et al attempts to understand how vaccination rates interact with changing population behavior to impact the course of the COVID-19 pandemic. They develop a novel mathematical model that incorporates behavioral changes into traditional age-structured dynamics, and compare pandemic trajectories across a wide-range of parameter values. Overall, their model presents an interesting finding that vaccination can actually cause pandemic surges if populations relax too quickly alongside the rollout. While I believe this to be an interesting and important contribution to the field, I have a number of concerns that I believe should be addressed before publication.

Major Comments

Overall, I believe there is a bit of redundancy in the manuscript, and I would suggest that the authors work hard to reduce the word count and/or reduce the figure count. For example, figures 1 through 4 alongside their respective descriptions in the results present many of the same conclusions, but with slight differences. To improve clarity, I would suggest attempting to focus some of the results around the main conclusions with some of the intricacies added to the supplemental information. Perhaps, even, the manuscript could focus solely on exploring the calibrated model results, and the simulated results could be added to the supplement and summarized briefly.

While I understand that it is important to not needlessly develop overly complex mathematical models, I believe the authors have oversimplified the vaccination process, which may impact their overall conclusions. First, the authors only allow for susceptible people in their model to be vaccinated. Realistically, many vaccines go to people who have already been infected and are recovered. This could change the relative dynamics between countries with varying levels of infection-derived immunities. Second, as the authors note, there is strong evidence that the vaccines prevent severe outcomes from breakthrough infections. The current model structure treats vaccine breakthrough infections similarly to traditional infections. Altering this to more accurately capture the vaccine protection could decrease the overall mortality rates in the simulations, but it might have larger impacts at different levels of alpha, so I think it would be worth investigating.

The authors model noncompliance rates of vaccinated and unvaccinated individuals with the same value, and don’t seem to explore scenarios where vaccinated individuals are more likely to be noncompliant than unvaccinated individuals. I think it would be interesting, and more realistic to think about scenarios where noncompliance rates differ between vaccinated and unvaccinated individuals, and suggest that the authors carry out a sensitivity analysis to understand the dynamics or provide more justification for that assumption in their model.

Given the novel model structure and implementation of noncompliance behavior, I think the authors should add a bit more information about the impact that the behavioral dynamics are having on their results. I believe this to be one of the most interesting aspects of the paper and is underexplored in the results. If the authors do choose to reduce the current figure/word count then I think it would make room for a figure and results exploring the impact the behavioral changes are having. A number of interesting questions could be explored (I am not suggesting exploring all of these, but they’re meant to be illustrative of some of the questions I had while reading): how do the behavioral dynamics impact epidemic curves? What fraction of the population is actually becoming noncompliant in the various simulations? Is there a specific relationship between vaccine efficacy and noncompliance rates that can help us understand the ultimate relative death difference? How do noncompliance rates within the model compare with the real-world?

Minor Comments

While I don’t think it will dramatically impact the results, I think it would be useful to have more justification for the assumption that everyone in an age group gets vaccinated. In most regions we are seeing saturation of vaccination rollout under 100%, so it will be useful to ensure the results hold for those scenarios.

One aspect that could be interesting to consider is the difference in countries in anticipated vaccination rollout plans. For example do most of the investigated countries have similar numbers of vaccines proportional to their population, or are there large differences between them?

The authors model the proportion of asymptomatic similarly across all age groups, but there are key differences between age groups in terms of asymptomatic rates. I would suggest changing this assumption to match the data, or justifying it more for the purposes of the study. See: https://www.nature.com/articles/s41591-020-0962-9

Since most countries are not homogeneously distributing vaccines, I thought it might be more realistic to have a different realistic scenario that is a middle ground between prioritizing the eldest age groups and the highest transmitting age groups: splitting vaccine prioritization between the high spreaders and high mortality age groups at the same time.

The expressions for g(alpha) and h(gamma) are not clear in the model. For example line 566 is different from line 514, and their relationships are unclear.

While I believe all of the assumed parameter values are provided for the model described in equation 1, I think it would be useful to have a supplementary table that outlines all of the parameters used in the model that are and are not fitted to data (the model calibration explanation of parameters is extremely clear for those that are fitted).

While the paper links to the contact matrices used for their age groups, I think it would be useful to outline what age groups are investigated in the model particularly because the vaccine prioritization strategies are age-based

Is there any justification for choosing r = 1.3 or r = 1.5? It might be useful to think about other forms of noncompliance (e.g. masking behavior), and the impact they had on transmission dynamics in justifying these values.

The authors don’t discuss the potential impact of testing on the ultimate epidemic trajectories. It might be interesting to consider the relative testing capacities of the countries, and what impacts they could have on the pandemic.

In lines 250 to 252, it is stated that: “Moreover, it is worth stressing that these results are to be intended in relative terms: a relative worst performance in averting deaths does not necessarily imply a worst absolute performance”. Could the authors please explain what they mean related to absolute performance. If there are important differences, I would suggest including a supplemental figure that demonstrates this dynamic.

Reviewer #2: In the manuscript entitled “The importance of non-pharmaceutical interventions during the COVID-19 vaccine rollout” Gozzi et al use an aged-structure mathematical model to explore how different key factors (vaccine efficacy, vaccination rate, vaccine allocation, behavioural changes) affect the impact of vaccine rollout in six different countries.

This is an interesting paper with interesting results. While there are a lot of others who have already reached similar conclusions (that NPIs need to be maintained through vaccine rollout) this paper is a nice addition to the existing literature because it attempts to explicitly model the influence of vaccination in human behavior and because it explores all these scenarios in 6 countries where little to no modeling has been done before.

Major concerns

1. While this modeling of human behavior is indeed attractive, it also is where my major concern is. The authors model this through a parameter alpha that ranges from 0 to 1000. This parameter modulates the fraction of susceptibles or vaccinated individuals moving to the classes with increased risk-behavior. There is no rationale or convincing argument as of why/how this parameter was chosen, or why the functional form using this parameter was chosen. The only reference given for this choice is yet another modeling paper, that was not based on data. In fact, the functional form for g(\\alpha) is not explicitly given in the paper. Furthermore, the authors state that people in those compartments will have an increased infection probability (r = 1.3 or 1.5). This corresponds to a 30% or 50% increase in transmission for these groups. Again, what is the rationale for these values? Are these values based in some knowledge?

The main conclusions of the paper, that with high values of alpha the impact of behavioral change might even produce more deaths, need to be more quantified and put in context with what these parameters are doing. For instance, for a rate of \\alpha > 10 and slow vaccination rate, there is a potential loss of 1.5.

What does alpha > 10 mean? It would be helpful to see what percentage of the compartments (susceptibles or vaccinated) is transitioning to the NC compartments with different values of \\alpha to see if these values of \\alpha make sense.

As the paper stands right now, it is not relatable to a wider audience or to public health officials. It would have much more impact if the authors can place their parameters in a context that is understandable by decision makers.

Is a loss of 1.5 equivalent to say that there is an increase of 50% more deaths compared to the non-vaccination scenario? This is unclear to me. If this is the correct interpretation of the numbers in Figs 1 and 2 it would be very helpful for the reader to put the results in these terms rather than potential losses (potential loss of 0.21 l228).

2. Overall, I think the paper would benefit of rewriting the results in a more organized and compact way. Right now, there are results in the results section, in the SM and even in the methods section (eg line 547)

The first part of the paper really deals with the problem of how changes VE, vaccination rates and behavior would affect vaccination campaigns for 6 countries with different population structures, with all the other parameters fixed. The important comparisons then should reside in trying to explain how the population structure is driving the results, and I believe that more can be said in that topic, as right now there is only a vague sentence trying to explain this (l 248). It might be good to plot which age group is moving to the NC compartments, and which age groups are dying in each country to try to figure out why countries with larger proportion of younger people feared worse.

The second part of the paper repeats this exercise with now fitted models to each country. I believe the results with the calibrated models are more interesting, and it might be worth it to put those first. Again, some interpretation of the values of \\alpha is really needed here.

3. The infection prevalence (number of people currently infected) set at 1% of the population is very high. To date, Canada’s worst day ~9,000 new cases. Even if one assumes that only 20% of infections are reported and that each day there are on average 5x the number of new cases, this comes back to ~ 0.0059 of the population (0.5%) infected.

Minor concerns:

1. Fig. 1 is too busy and hard to read. Part A can go to the supplement or split figure 1 into two figures. The middle panel of A is unclear. Does it mean that People in Peru aged 15-19 have >100 more contacts than people in Italy of the same age group? I wonder if there is a better representation of this. Maybe a barplot with the number of contacts for each age group for all 6 countries (one could split it into 2 rows with 8 age groups per row) Part B is too small to note any differences in strategies.

2. l.227 “strategy reducing severity” suggest putting in parenthesis the strategy (strategy 1).

3. The paper would benefit with a table with the parameters used in the model and a supplemental table with the values obtained by fitting (given in fig4A)

4. Figure 5 is not at all clear to me, not clear how the point it is trying to convey is actually conveyed with the figure.

5. While the authors cite some of the first papers done in vaccine allocation for COVID-19, there is a vast literature now of papers that addressed very similar issues to what they are addressing ( e.g.

https://academic.oup.com/cid/advance-article/doi/10.1093/cid/ciab079/6124429, https://www.medrxiv.org/content/10.1101/2020.12.31.20249099v4, https://www.thelancet.com/journals/laninf/article/PIIS1473-3099(21)00143-2/fulltext

https://www.medrxiv.org/content/10.1101/2020.12.30.20248888v2 just to name a few). Their results need to be put in context.

**Have the authors made all data and (if applicable) computational code underlying the findings in their manuscript fully available?**

Reviewer #1: **No: **I didn't see any link to the code for the analysis.

Reviewer #2: **No: **Statement regarding code not found.

PLOS authors have the option to publish the peer review history of their article (what does this mean?). If published, this will include your full peer review and any attached files.

Reviewer #1: No

Reviewer #2: No
---

## [Decision Letter · Decision Letter 1]

12 Aug 2021

Dear Dr. Bajardi,

We are pleased to inform you that your manuscript 'The importance of non-pharmaceutical interventions during the COVID-19 vaccine rollout' has been provisionally accepted for publication in PLOS Computational Biology.

Best regards,

Roger Dimitri Kouyos

Associate Editor

PLOS Computational Biology

Virginia Pitzer

Deputy Editor-in-Chief

PLOS Computational Biology

Reviewer's Responses to Questions

**Comments to the Authors:**

Reviewer #1: I thank the authors for taking such time and effort on my suggested reviews. I have no further comments and believe the paper is of sufficient quality to warrant publication.

Reviewer #2: This manuscript has been greatly improved!

My only comment is to use the same value of alpha for figures 2 (alpha=1) and 3 (alpha =3). The value of alpha used for figure 3 seems a little bit unrealistic to me: Vaccinated people will all be non-compliant almost immediately. I suggest using a lower value of alpha or to mention everywhere this value is used that this really is a "worse case scenario".

**Have the authors made all data and (if applicable) computational code underlying the findings in their manuscript fully available?**

Reviewer #1: None

Reviewer #2: Yes

PLOS authors have the option to publish the peer review history of their article (what does this mean?). If published, this will include your full peer review and any attached files.

Reviewer #1: No

Reviewer #2: **Yes: **Laura Matrajt

---

## [Editor Report · Acceptance letter]

3 Sep 2021

PCOMPBIOL-D-21-00697R1 

The importance of non-pharmaceutical interventions during the COVID-19 vaccine rollout

Dear Dr Bajardi,

I am pleased to inform you that your manuscript has been formally accepted for publication in PLOS Computational Biology. Your manuscript is now with our production department and you will be notified of the publication date in due course.

With kind regards,

Amy Kiss
